# The accuracy of absolute differential abundance analysis from relative count data

**Kimberly E. Roche** [1] *, **Sayan Mukherjee** [1,2,3,4]

**1** Program in Computational Biology and Bioinformatics, Duke University, Durham, North Carolina, United States of America, **2** Departments of Statistical Science, Mathematics, Computer Science, Biostatistics & Bioinformatics, Duke University, Durham, North Carolina, United States of America, **3** Center for Scalable Data Analytics and Artificial Intelligence, Universität Leipzig and the Max Planck Institute for Mathematics in the Natural Sciences, Leipzig, Germany, **4** Center for Genomic and Computational Biology, Duke University, Durham, North Carolina, United States of America

* kimberly.roche@duke.edu

**Data Availability Statement:** For previously published studies, sequencing data were generally obtained from GEO repositories: GSE161116 (Song et al.), GSE107011 (Monaco et al.), GSE54695 (Gruen et al.), GSE85241 (Muraro

## Abstract

Concerns have been raised about the use of relative abundance data derived from next generation sequencing as a proxy for absolute abundances. For example, in the differential abundance setting, compositional effects in relative abundance data may give rise to spurious differences (false positives) when considered from the absolute perspective. In practice however, relative abundances are often transformed by renormalization strategies intended to compensate for these effects and the scope of the practical problem remains unclear. We used simulated data to explore the consistency of differential abundance calling on renormalized relative abundances versus absolute abundances and find that, while overall consistency is high, with a median sensitivity (true positive rates) of 0.91 and specificity (1— false positive rates) of 0.89, consistency can be much lower where there is widespread change in the abundance of features across conditions. We confirm these findings on a large number of real data sets drawn from 16S metabarcoding, expression array, bulk RNA-seq, and single-cell RNA-seq experiments, where data sets with the greatest change between experimental conditions are also those with the highest false positive rates. Finally, we evaluate the predictive utility of summary features of relative abundance data themselves. Estimates of sparsity and the prevalence of feature-level change in relative abundance data give reasonable predictions of discrepancy in differential abundance calling in simulated data and can provide useful bounds for worst-case outcomes in real data.

## Author summary

Molecular sequence counting is a near-ubituiqous method for taking "snapshots" of the state of biological systems at the molecular level and is applied to problems as diverse as profiling gene expression and characterizing bacterial community composition. However, concerns exist about the interpretation of these data, given they are relative counts. In particular some feature-level differences between samples may be technical, not biological, stemming from compositional effects. Here, we quantify the accuracy of estimates of

et al.), GSE121655 (Kimmerling et al.), GSE65525 (Klein et al.), GSE53960 (Yu et al.), and GSE65785 (Owens et al.). Sequencing data from the Hashimshony et al. CEL-Seq2 study were obtained from GEO via accession code GSE78779 and subset to GFP-negative (quiescent) and GFP-positive (cycling) fibroblast samples only. Data from Hagai et al. were downloaded from ArrayExpress via accession E-MTAB-6754. Only the subset of mouse samples for unstimulated and 4-hour time points were used. The quantitative microbiome profiling data from Vieira-Silva et al. were obtained from the author's website at: http://raeslab.org/software/QMP2/. Finally, the digital droplet PCR-normalized data from Barlow et al. was obtained from the "Source Data" supplement of the published article with DOI: https://doi.org/10.1038/s41467-020-16224-6. Original data and code for this study are available in the public Github repository located at https://github.com/kimberlyroche/codaDE.

**Funding:** The authors would like to acknowledge funding from Human Frontier Science Program grant HFSP RGP005 (to SM), National Science Foundation grants NSF DMS 17-13012 (to SM), NSF BCS 1552848 (to SM), NSF DBI 1661386 (to SM), NSF IIS 15-46331 (to SM), and NSF DMS 16-13261 (to SM), as well as funding from the North Carolina Biotechnology Center through grants 2016-IDG-1013 (to KR, SM) and 2020-IIG-2109 (to KR, SM). The authors would also like to acknowledge funding through a graduate fellowship provided by the Duke Forge health data science center (to KR). No funders played any role in the study design, data collection and analysis, decision to publish, or preparation of the manuscript.

**Competing interests:** The authors have declared that no competing interests exist.

sample-sample differences made from relative versus "absolute" molecular count data, using a comprehensive simulation strategy and published experimental data. We find the accuracy of difference estimation is high in at least 50% of simulated and real data sets but that low accuracy outcomes are far from rare. Further, we observe similar numbers of these low accuracy cases when using any of several popular methods for estimating differences in biological count data. Our results support the use of complementary reference measures of absolute abundance (like RNA spike-ins) for normalizing next-generation sequencing data. We briefly validate the use of these reference quantities and of stringent effect size thresholds as strategies for mitigating interpretational problems with relative count data.

This is a *PLOS Computational Biology* Methods paper.

## Introduction

Warnings about the consequences of compositional effects in sequence count data have been published repeatedly in the decades since the technology's advent and its application to a host of biological problems. The issue relates to a loss of scale information during sample processing, which renders counts of genes, transcripts, or bacterial species as relative abundances. The technical basis for this belief is summarized in Box 1. No consensus solution for this problem exists. In this work, we use prominent differential abundance testing methods on simulated and real data to quantify the discrepancy between differential abundance estimates made on relative versus "absolute" abundances. Our simulations show that methods which heuristically rescale sample abundances are often reasonably consistent in their estimates of change across relative and absolute counts but that, in a sizeable number of cases characterized by substantial change across simulated contiditions, most or all methods differential abundance testing methods evaluated yielded inaccurate estimates. Further, we show that data sets which are

### Box 1: Measuring *relative* abundances

Sequence counting has become widespread as a means of census-taking in microscopic biological systems. Genomic material, typically RNA, is captured and quantified at the component level. Sampled cells are lysed, messenger RNA is captured and fragmented, transcribed into cDNA, sequenced, classified, and quantified. The results are relative abundances of gene products in the cell (in the case of single-cell RNA-seq) or tissue (in bulk RNA-seq). In another instance, whole bacterial communities are profiled by bar-coding of the 16S subunit of the ribosome. Ribosomal RNA associated with this piece of translation machinery is ubiquitously present across the bacterial kingdom but variations in the genetic sequence of this component can uniquely identify bacteria to the species or strain level in well-characterized systems, allowing a researcher to profile bacterial community composition. Absent measurements of microbial load or transcriptome size, however, the observed sequence counts in all these cases represent relative abundances.

Sequence count data is compositional due to steps in sample processing. Across domains, samples are typically normalized to some optimal total amount of genetic material prior to sequencing in accordance with manufacturer recommendations for best performance. This step removes variation in total abundance across samples.

Saturation of sequencing has been cited [18] as another mechanism by which abundances are rendered relative: a finite amount of reagent means there is an upper limit on biological material which can be captured; rare components can be forced out by a "competition" to be sampled. These factors withstanding, observed total abundances would likely still be noisy. Repeated subsampling of small amounts of material and variation in the efficiency of library preparation steps can distort observed totals.

In transcriptomics and in microbial community profiling, residual variation in observed total abundances across samples is generally regarded as a source of technical bias and most analytics pipelines involve steps to rescale observed abundances. The simplest of these is the counts per million (CPM) transformation which converts observed counts to relative abundances, then scales by 1 million.

especially susceptible to distortion by compositional effects can sometimes be predicted on the basis of "signatures" of this distortion.

## Compositionality in sequence count data

Compositionality refers to the nature of sequence count data as containing relative abundance information only. In the differential abundance setting, several authors [1–3] have described the problem this poses: whereas researchers would like to interpret change in absolute abundances, compositional effects mean using change in relative abundances as a proxy can lead to false discoveries. A few authors have cited instances of these false discoveries in real data. Coate and Doyle [4, 5] discussed the issue of transcriptome size variation in plants and other systems and the impact of this on accurate transcriptome profiling. Nie *et al* and Lin *et al* [6, 7] documented the phenomenon of widespread "transcription amplification" by the transcription factor *c-Myc* and Lovén *et al* [8] used *c-Myc* data and parallel RNA quantification assays to show that substantial differences in total abundance between control and elevated *c-Myc* conditions resulted in very different interpretations of apparent differential expression.

Common to these studies of transcriptomes is a recommendation that, where feasible, researchers leverage RNA spike-ins as controls against which changes in observed abundance can be scaled [9, 10]. But this practice has fallen short of widespread adoption. While several papers have expressed confidence in the utility of spike-ins [11–15], the doubt cast by reports of widespread batch effects [16] and technical noise [17] have had the effect of reducing researcher confidence in their use. Further, the introduction of spike-ins is not practical on all platforms.

Where approaches that rely on spike-ins are undesirable or infeasible, sample rescaling procedures have proliferated. These methods typically assume the existence of a stable set of features and attempt to normalize compositions in such a way as to recover this stable set across samples. In fact, in transcriptomics, these methods predominate.

In the setting of microbial community profiling, the prevailing assumption is that typical compositions are too simple for rescaling methods to work well (although results in benchmarking studies have been mixed [19, 20]). Competing approaches have been developed for dealing with compositionality in microbial sequence count data. Quantitative microbiome profiling [21] and similar approaches combine relative abundances with complementary measurements of microbial load to reconstruct absolute abundances. In contrast, so-called compositional methods are also utilized. These involve log relative representations which can give approximate log-normality, such that workhorse statistical methods for continuous data may

be applied. However, interpretation of these quantities can be challenging (e.g. as with the isometric logratio [22]).

Though there is evidence from simulated and real data that *scale*—i.e. increasing complexity of composition in terms of numbers of genes, transcripts, or bacterial sequence variants—mitigates the problem of compositionality [2, 20], it remains unclear whether there are instances where it is reasonable to substitute relative abundances for absolute abundances and several fields could benefit from clarity on this issue. In this work, we quantify the discrepancy in differential abundance calling on simulated and real data sets representative of 16S metabarcoding, bulk RNA-seq, and single-cell RNA-seq experiments.

In particular, we evaluate a set of methods which rescale observed total abundances and show that all such rescaling strategies outperform a simple library size normalization in our simulated data. The methods selected are a diverse set meant to be representative but not exhaustive and draw from the fields of transcriptomic and metagenomic analysis: ALDEx2 [23], ANCOM-BC [24], DESeq2 [25], edgeR [26], and scran [27]. Each of these methods rescales the observed counts against a reference quantity, implicit or explicit—typically, a subset of putatively stable features. Where such a stable reference exists, differences in these rescaled counts should approximate differences in absolute counts. In all cases, baseline or "true" differential abundance for each feature in the absolute count data was determined by the use of a simple generalized linear model. This yielded a common standard against which to compare differential abundance calls made by ALDEx2, ANCOM-BC, DESeq2, edgeR, and scran. Details on the simulation procedure and these differential abundance calling algorithms are given in Methods.

In simulations exploring a broad range of differential abundance scenarios, we find median false positive rates of differential abundance calls made on absolute versus relative counts are 11.5% and that false positive counts increase with the proportion of differentially abundant features. The number of features (e.g. genes or bacterial sequence variants) plays little role in observed outcomes. An exploration of sequencing data collected from twelve external studies reveals similar trends. Further, we show that summaries of sparsity and the prevalence of apparent feature-level change can provide reasonable predictions of the amount of discrepancy in differential abundance calls in simulated data and may be useful for bounding expectations of discrepancy in real data sets.

## Results

We simulated differentially abundant count data in paired sets of absolute and relative abundances. In the absolute abundances, differentially abundant features experienced either an increase or decrease in abundance—often large—between each of two simulated conditions. Large numbers of differentially abundant features frequently had the effect of changing the overall total abundance, potentially resulting in several-fold changes in scale between conditions. In each simulated data set a subset of randomly selected features received a random increase or decrease in average abundance between simulated conditions, contributing to an overall increase or decrease in total abundance which might be large or small. These changes in total abundance were purposely removed by a fixed-depth resampling to give a set of relative abundances. The proportion of differentially abundant features simulated and other key characteristics of the simulated data are summarized in S1 and S2 Figs. Compositional effects might be present in this relative count data as *relative* change exhibited by non-differentially abundant features.

We explored ranges of feature number and in the amount of differential abundance, grouping simulations into three partially overlapping settings: a **Microbial** setting, characterized by low feature number and high differential abundance; a **Bulk Transcriptomic** setting having

high feature number and low differential abundance; and an intermediate **Cell Transcriptomic** setting. The full results from 5625 simulations are summarized in S1 Table and visualized in Fig 1, where agreement between differential abundance calls on absolute and relative abundances for each data set are summarized by means of sensitivity and specificity statistics, as described below. The same results are labeled in terms of increasing simulated fold change in S3 Fig and organized by increasing feature number in S4 Fig. In all cases, differential features were called with FDR $\leq$ 0.05 on Benjamini-Hochberg adjusted p-values obtained from each of the methods of interest.

We report outcomes in terms of sensitivity (true positive rate) and specificity (1—false positive rate). Perfect concordance of differential abundance calls made on observed versus absolute counts would yield a sensitivity of 1.0 and a specificity of 1.0. Sensitivity drops as more features deemed differentially abundant from the perspective of the absolute counts fail to appear significantly different in the observed data and specificity drops as an increasing number of features appear differentially abundant in the observed counts alone. We highlight key observations made on simulated data below.

## Simulated data

**All rescaling methods have higher accuracy than edgeR with library size normalization.**   Library size—or total count—normalization represented an anticipated worst case for accuracy in differential abundance testing. In order to test this, we employed edgeR, which allows for either total count or trimmed mean of M-values (TMM) normalization [26]. The difference in accuracy outcomes with and without TMM normalization was substantial. Median specificity was only 0.35 when the observed per-sample totals were normalized by total counts, versus 0.88 when using TMM normalization on the same collection of simulated data sets. False positive counts were over four times higher in the absence of TMM normalization, demonstrating that, for edgeR, a very widely used differential abundance calling method, accuracy is greatly improved by this heuristic per-sample rescaling. These results are illustrated in Fig 2.

**Moderate specificity was typical but cases of low specificity were observed in all settings for all methods.**   Median specificity was moderately high at 0.885—lowest for ANCOM-BC in the Cell Transcriptomic setting at 0.835 and highest for ALDEx2 in the same setting at 0.937 (see Table 1). However, a minority of simulated data sets yielding very large false positive rates were observed in every setting. Data sets with very low specificity ($<$ 0.5) made up less than 10% of simulated cases in the Transcriptomic settings and more than 14% in the Microbial setting.

**Increasing feature number improves specificity but the effect is modest.**   Median specificity improved slightly for ALDEx2 and scran in the highest-feature Bulk Transcriptomic setting relative to the lowest-feature Microbial setting. The change in false positive counts in particular is visualized in Fig 2. For all other methods, accuracy remained similar across settings.

**Though scran was the top performer, outcomes were similar across methods.**   Scran had the highest overall accuracy with a median sensitivity of 0.93 and specificity of 0.91. Sensitivity was lower in ALDEx2 (median = 0.78) and specificity was lower in DESeq2 (median = 0.86). All in all, methods exhibited similar performance in terms of their distributions of sensitivity and specificity across settings. See Figs 1 and 2 and Tables 1 and 2.

Next, we explored the characteristics of relative abundances associated—either positively or negatively—with observed sensitivities and specificities in simulated data. It is possible to imagine characteristics which might indicate the presence of distortion by compositional effects, for example, an increase in the percent of rare features from one condition to the next

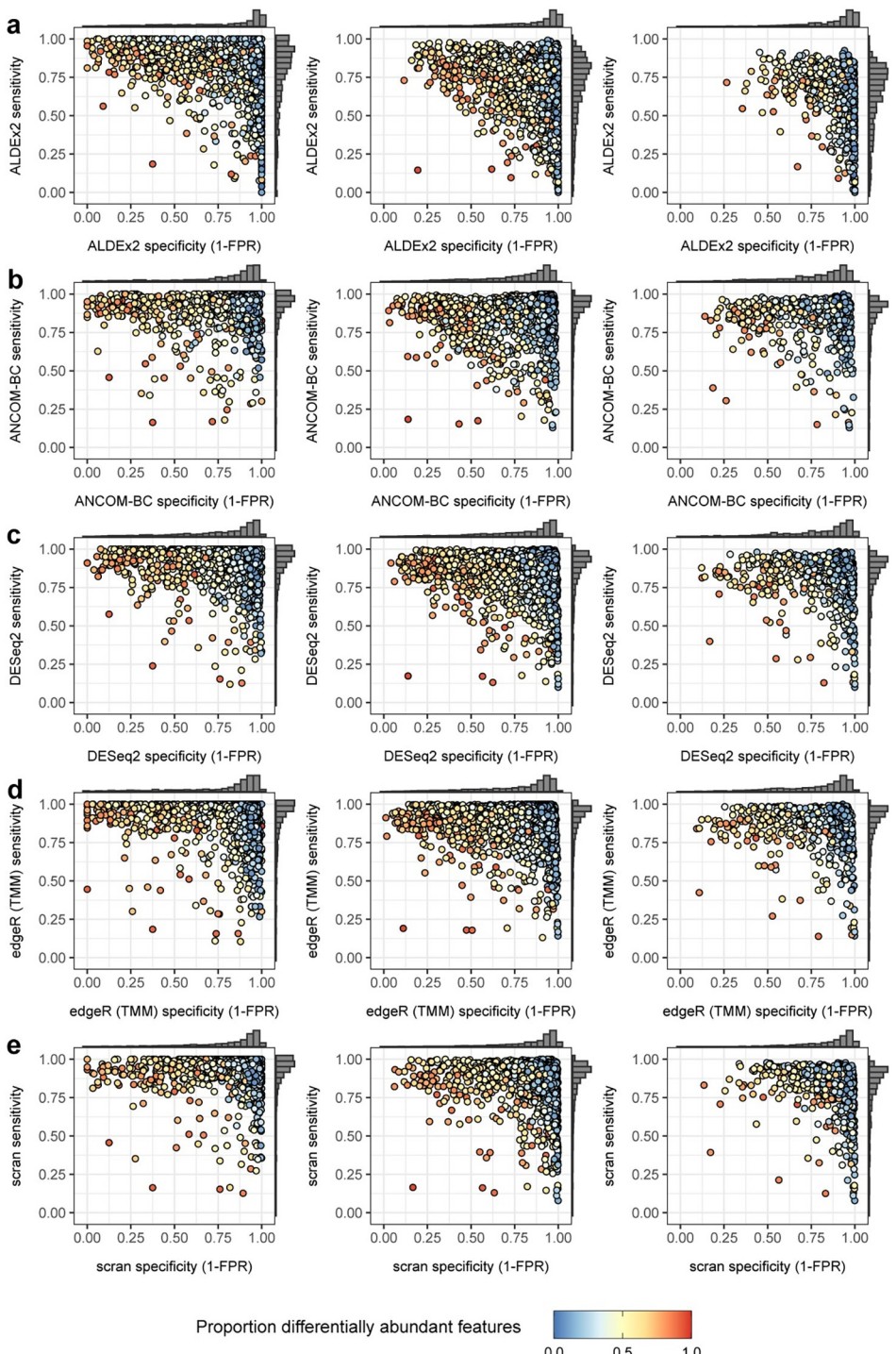

**Fig 1. Simulated sensitivity and specificity as a function of setting and proportion of differentially abundant features.** Sensitivity and specificity for five differential abundance calling methods in three experimental settings: **a)** Microbial, **b)** Bulk Transcriptomic, and **c)** Cell Transcriptomic settings. Data sets are labeled by proportion of differentially abundant features.

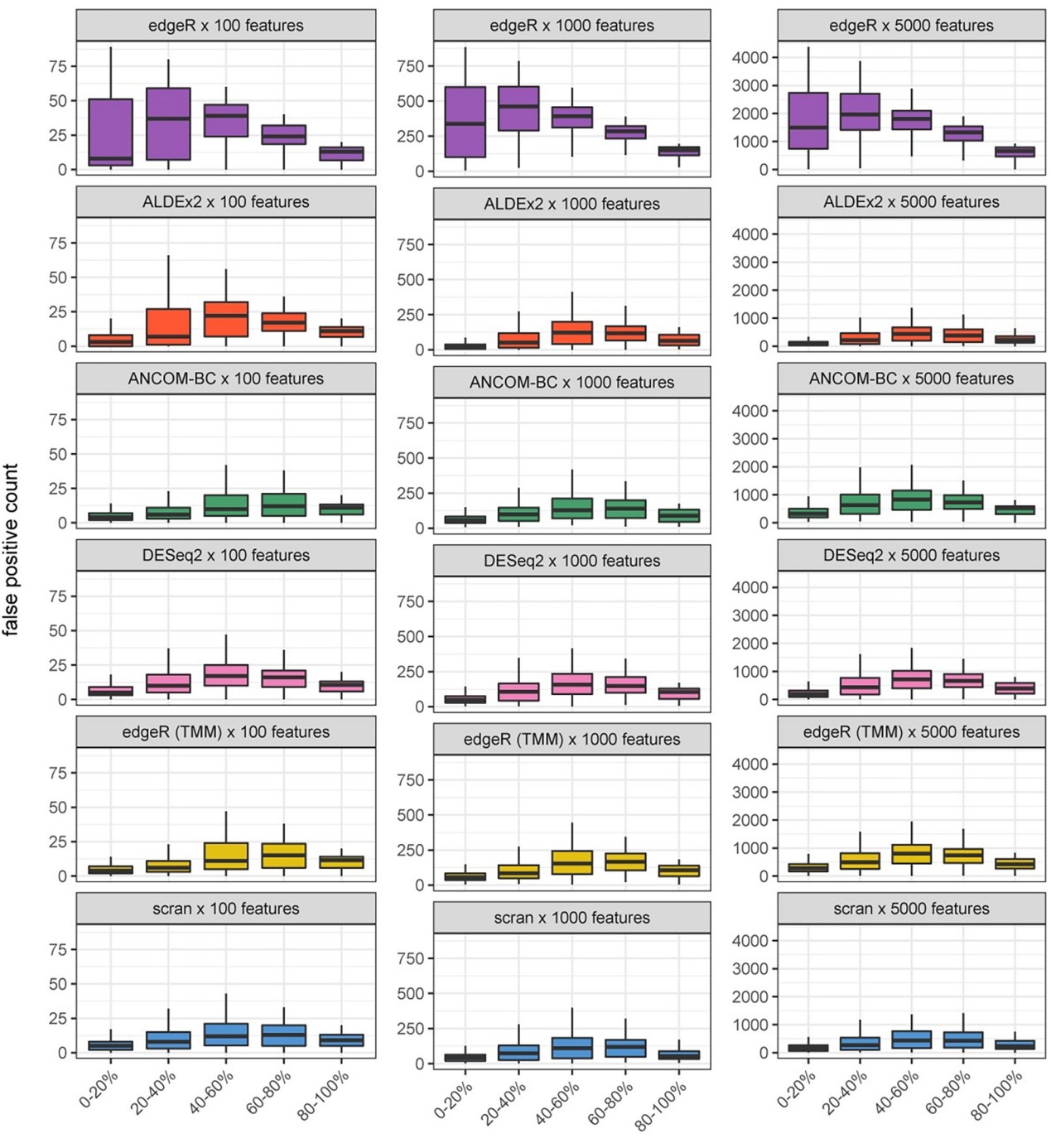

**Fig 2. False positive counts as a function of percent differentially abundant features in simulation.** Counts of false positives as a function of the percent of simulated differentially abundant features for six methods. edgeR with no heuristic rescaling is shown at top. Columns segregate simulations with increasing numbers of features. All rescaling methods have higher overall accuracy than edgeR without rescaling. Counts of false positives tend to be highest where a majority of features are differentially abundant.

**Table 1. Median and thresholded specificity by simulated setting, FDR ≤ 0.05.** Low specificity is a common outcome in simulated data. A majority of simulated data sets have false positive rates in excess of 5%. A minority of data sets have false positive rates in excess of 50%.

| Setting | Method | Median specificity | Percent of data sets below 95% specificity | Percent of data sets below 50% specificity |
|---|---|---|---|---|
| Microbial | ALDEx2 | 0.915 | 57% | 18% |
| Microbial | ANCOM-BC | 0.886 | 78% | 13% |
| Microbial | DESeq2 | 0.856 | 78% | 15% |
| Microbial | edgeR (TMM) | 0.887 | 78% | 16% |
| Microbial | scran | 0.901 | 71% | 9% |
| Cell Transcriptomic | ALDEx2 | 0.937 | 56% | 5% |
| Cell Transcriptomic | ANCOM-BC | 0.835 | 88% | 14% |
| Cell Transcriptomic | DESeq2 | 0.865 | 76% | 11% |
| Cell Transcriptomic | edgeR (TMM) | 0.858 | 84% | 13% |
| Cell Transcriptomic | scran | 0.917 | 67% | 6% |
| Bulk Transcriptomic | ALDEx2 | 0.935 | 56% | 2% |
| Bulk Transcriptomic | ANCOM-BC | 0.853 | 87% | 11% |
| Bulk Transcriptomic | DESeq2 | 0.865 | 77% | 10% |
| Bulk Transcriptomic | edgeR (TMM) | 0.861 | 85% | 12% |
| Bulk Transcriptomic | scran | 0.912 | 66% | 4% |

(in effect, *dropouts*). While we might not expect any single such characteristic to predict the sensitivity or specificity of calls made on relative versus absolute abundances, composites of such characteristics might be informative. We describe the strongest associations we observed below.

**Sensitivity is anti-correlated with estimates of sparsity.** Large proportions of zero- and one-counts in the simulated relative abundances correlated with low sensitivity (Spearman's correlation between proportion zeros and sensitivity, $\rho = -0.58$). In effect, this echoes similar findings that decreasing sequencing depth decreases power in genomics studies [14] and reflects a lesser expected statistical confidence in the observed change of low-count features.

**Table 2. Median and thresholded specificity by simulated setting, FDR ≤ 0.01 plus effect size thresholding.** Specificity in simulated data under more stringent conditions: here differential abundant is called using FDR ≤ 0.01 and an effect size threshold (a fold change of at least 2-fold).

| Setting | Method | Median specificity | Percent of data sets below 95% specificity | Percent of data sets below 50% specificity |
|---|---|---|---|---|
| Microbial | ALDEx2 | 0.93 | 56% | 12% |
| Microbial | ANCOM-BC | 0.942 | 55% | 3% |
| Microbial | DESeq2 | 0.957 | 45% | 2% |
| Microbial | edgeR (TMM) | 0.945 | 53% | 3% |
| Microbial | scran | 0.961 | 42% | 2% |
| Cell Transcriptomic | ALDEx2 | 0.941 | 55% | 3% |
| Cell Transcriptomic | ANCOM-BC | 0.931 | 59% | 1% |
| Cell Transcriptomic | DESeq2 | 0.956 | 46% | 2% |
| Cell Transcriptomic | edgeR (TMM) | 0.943 | 55% | 2% |
| Cell Transcriptomic | scran | 0.964 | 40% | 1% |
| Bulk Transcriptomic | ALDEx2 | 0.936 | 56% | 0% |
| Bulk Transcriptomic | ANCOM-BC | 0.943 | 56% | 1% |
| Bulk Transcriptomic | DESeq2 | 0.958 | 46% | 2% |
| Bulk Transcriptomic | edgeR (TMM) | 0.944 | 54% | 2% |
| Bulk transcriptomic | scran | 0.964 | 40% | 1% |

Interestingly, in our simulations, evidence of extremes in terms of the apparent correlation of relative abundances (in fact, the skew of that distribution of correlations) was also inversely associated with sensitivity ($\rho = -0.50$).

**Specificity is strongly anti-correlated with the estimated proportion of differential features.** The proportions of features undergoing large fold decreases or large fold increases in abundance relative to the mean were highly informative with respect to specificity ($\rho = -0.61$ and $\rho = -0.43$ respectively). The standard deviation of the change in log counts between simulated conditions was also anti-correlated with specificity ($\rho = -0.56$). In each case, these characteristics supplied evidence of the existence (or lack) of widespread change in composition. Methods which rescale observed abundances rely on a pool of stable reference features against which to estimate per-sample "scaling factors." The larger the number of components in the composition apparently changing, the greater the extent to which this rescaling is impaired.

We utilized these and over 50 additional features derived from relative abundance data to train per-method models of both sensitivity and specificity, with the aim of predicting the discrepancy in relative versus absolute differential abundance calls from features of observed data alone, reasoning that these "signatures" might be highly informative. All model features are outlined in S2 Table and feature importance is explored in S3 and S4 Tables. Predictive models were trained on 80% of our simulated data and their performance was evaluated on the held-out 20% of simulated data sets. Predictions of specificity were more accurate than predictions of sensitivity (mean specificity $R^2 = 0.70$; mean sensitivity $R^2 = 0.59$). Prediction of outcomes was most successful for ALDEx2, where the $R^2$ values for sensitivities and specificities on held-out data were 0.74 and 0.72, respectively. Full results are given in Table 3. These results suggested it might be reasonable to predict discrepancy between differential abundance calls made on absolute versus relative abundances using characteristics of the relative abundance data alone. However, the study of some representative real data sets will suggest prediction may be much more difficult in this setting.

## Real data

Next, we examined a variety of real data sets across many experimental settings in order to explore outcomes in real data. We collected publicly available data from twelve studies [28–39] and attempted to reconstruct absolute abundances by normalizing observed total sample abundances against reference quantities provided in the same published materials. In most cases, these reference quantities were external RNA spike-in sequences. In others, reconstructed absolute abundances had already been estimated, as in [28] through quantitative microbiome profiling (QMP [21]). In one case [38], we normalized against the housekeeping gene *gapdh* [40] and in another [33], against paired measurements of cell mass. Visual and textual summaries of these data sets are available in Figs 3 and 4 and Table 4. While not exhaustive, these studies illustrate a wide range of experimental conditions. We acknowledge the

**Table 3. Predictive model performance summaries.** Performance of per-method random forest predictive models of sensitivity and specificity on held out data sets.

| Method | Sensitivity prediction $R^2$ | Specificity prediction $R^2$ |
|---|---|---|
| ALDEx2 | 0.737 | 0.721 |
| ANCOM-BC | 0.671 | 0.482 |
| DESeq2 | 0.517 | 0.819 |
| edgeR | 0.504 | 0.776 |
| scran | 0.526 | 0.699 |

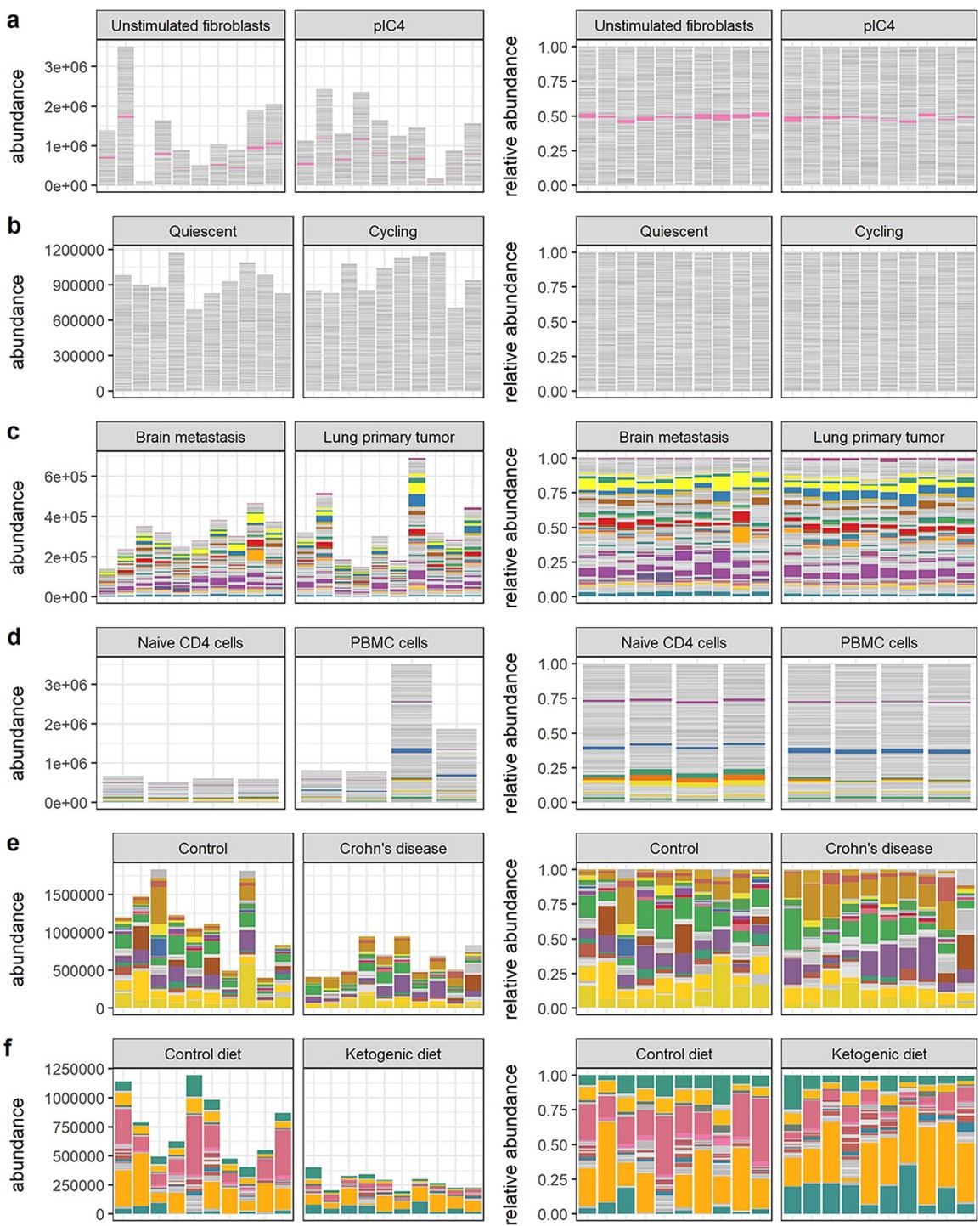

**Fig 3. Visual summaries of real data sets 1–6.** Visual summaries of nominal abundances (left panels) and relative abundances (right panels) for **a)** Hagai et al. **b)** Hashimshony et al. **c)** Song et al. **d)** Monaco et al. **e)** Vieira-Silva et al. **f)** Barlow et al. Features (genes or bacterial sequence variants) with at least 1% relative abundance across all samples are colored; all other features are gray.

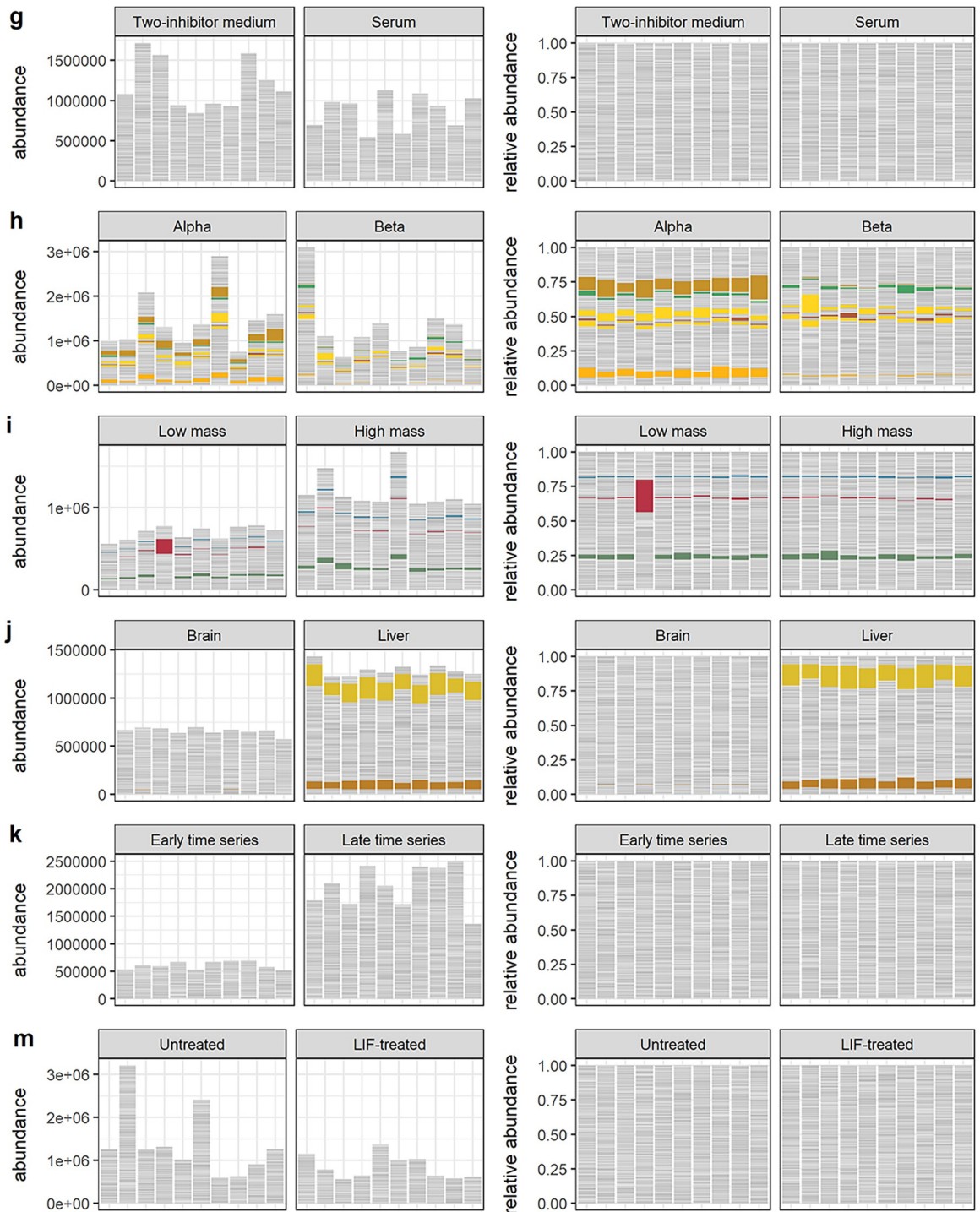

**Fig 4. Visual summaries of real data sets 7–12.** Visual summaries of nominal abundances (left panels) and relative abundances (right panels) for **g)** Grün et al. **h)** Muraro et al. **i)** Kimmerling et al. **j)** Yu et al. **k)** Owens et al. **m)** Klein et al. Features (genes or bacterial sequence variants) with at least 1% relative abundance across all samples are colored; all other features are gray.

**Table 4. Overview of real data sets.** Real 16S metabarcoding, bulk RNA-seq, and single cell RNA-seq data sets corresponding to the abundances shown in Figs 3 and 4. The estimated percent differential features are those significantly differential to a negative binomial GLM in the nominal abundances.

| Source | Description | Number sequence variants | Number samples (per-condition) | Percent zeros | Approx. fold change | Approx. percent differential features |
|---|---|---|---|---|---|---|
| Hagai et al. [30] | bulk RNA sequencing of both unstimulated and mock-viral infected mouse fibroblasts | 13937 | 10, 29 | 34% | 1.5 | 18% |
| Hashimshony et al. [31] | single cell RNA-sequencing of quiescent and cycling mouse fibroblasts | 9381 | 31, 38 | 9% | 1.3 | 32% |
| Song et al. [34] | nCounter array of human primary lung cancer vs. brain metastases | 765 | 13, 15 | 0% | 1.3 | 37% |
| Monaco et al. [36] | immune cell profiling in human humans via bulk RNA-seq | 20675 | 4, 8 | 15% | 3.4 | 45% |
| Vieira-Silva et al. [28] | 16S metagenomics from human gut samples of control and Crohn's disease patients | 76 | 14, 54 | 37% | 1.8 | 47% |
| Barlow et al. [35] | 16S metagenomics from ketogenic diet and control mice | 104 | 17, 18 | 89% | 3.4 | 47% |
| Grün et al. [32] | mouse embryonic stem cells cultured in serum and a two-inhibitor solution | 4652 | 76, 56 | 22% | 1.1 | 51% |
| Muraro et al. [29] | single cell RNA-seq of pancreatic islet cells | 3494 | 100, 100 | 37% | 1.1 | 59% |
| Kimmerling et al. [33] | cycling, stimulated CD8 + T cells | 9918 | 79, 79 | 35% | 2 | 74% |
| Yu et al. [37] | single cell expression profiling of rat brain and liver tissue | 26898 | 32, 32 | 15% | 2 | 79% |
| Owens et al. [39] | single cell sequencing of zebrafish embryos; early vs. late time course samples drawn | 40476 | 24, 35 | 16% | 3.7 | 89% |
| Klein et al. [38] | single cell RNA-sequencing of normally developing and leukemia inhibitory factor-treated mouse ESCs | 2928 | 100, 100 | 37% | 2.9 | 98% |

difficulty in reconstructing absolute abundances and caution that these estimates are likely noisy. Rather than call them absolute abundances, we will adopt the helpful terminology of [9] and call these reconstructions "nominal abundances" to distinguish them from some theoretical ground truth. While these quantities are an abstraction, since they derive from real experiments, we believe they capture some of the real variation in composition and scale we could expect to see in data from typical experiments and should serve to ground expectations. As with simulated data, we proceed by comparing differential abundances called on the relative count data (via ALDEx2, ANCOM-BC, DESeq2, edgeR, or scran) with differential abundances called on the nominal abundances—our proxy for "true" differential abundance.

We have endeavored to include among the real data sets some of the most challenging possible cases for differential abundance calling from relative count data. Many of the data sets we have selected involve a large amount of absolute and/or compositional change across experimental or biological conditions. These data sets help us answer the questions: What is the scale of discrepancy in the worst cases? How useful are summaries derived from relative abundances in these cases? Could a researcher reasonably predict error from relative abundance data alone?

Accuracy outcomes for these real data sets are given in Figs 5 and 6 and S5 and S6 Tables. These outcomes can be broadly classified on the basis of sensitivity into low, moderate, and high sensitivity cases, described below.

**Low sensitivity, high specificity cases.** Three data sets exhibited a combination of very low sensitivity (median = 0.06) and high specificity (median = 0.99). These were the data of

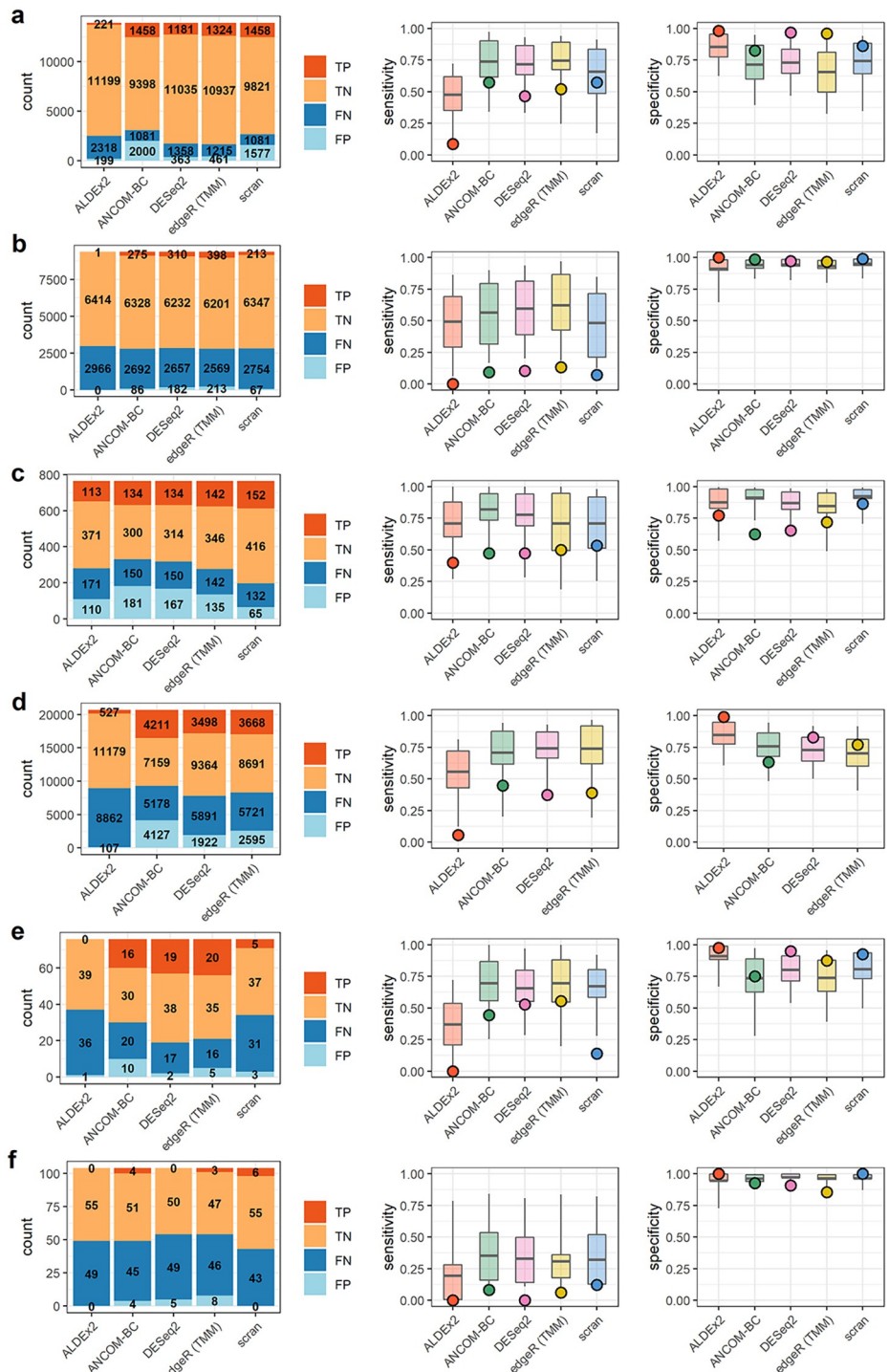

**Fig 5. Results for real data sets 1–6.** Results summaries for **a)** Hagai et al. **b)** Hashimshony et al. **c)** Song et al. **d)** Monaco et al. **e)** Vieira-Silva et al. **f)** Barlow et al. Barplots on the left indicate counts of true and false positives (TP, FP) and true and false negatives (TN, FN). Boxplots on the right indicate the 50% (box) and 90% (line) intervals of prediction and points, the observed values of either sensitivity or specificity.

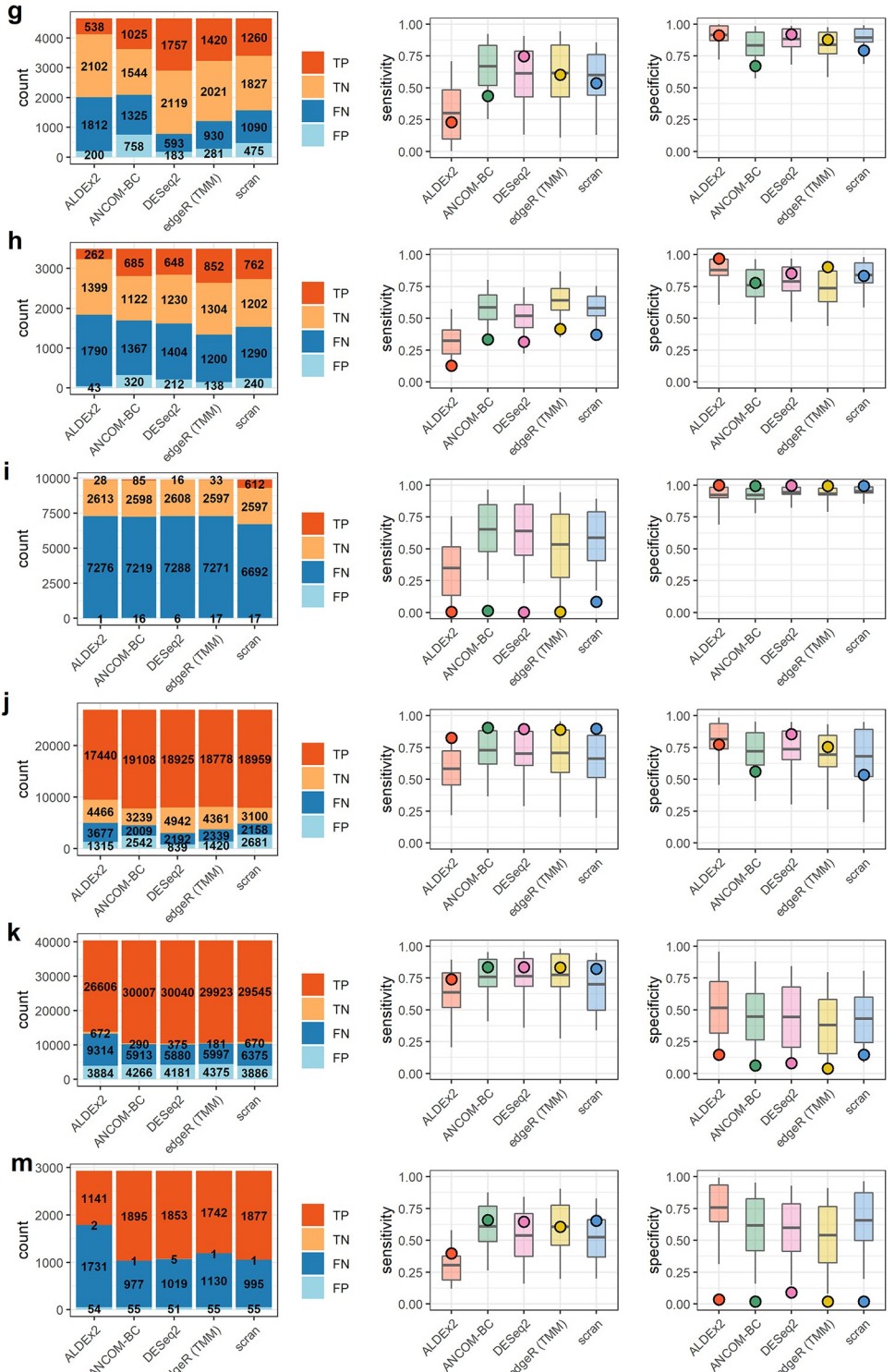

**Fig 6. Results for real data sets 7–12.** Results summaries for **g)** Grün et al. **h)** Muraro et al. **i)** Kimmerling et al. **j)** Yu et al. **k)** Owens et al. **m)** Klein et al. Barplots on the left indicate counts of true and false positives (TP, FP) and true and false negatives (TN, FN). Boxplots on the right indicate the 50% (box) and 90% (line) intervals of prediction and points, the observed values of either sensitivity or specificity.

Hashimshony et al., Barlow et al., and Kimmerling et al. (Figs 5 and 6). These studies consisted of two transcriptomic experiments—in human and mouse cells—and a single gut microbial data set. The range of differentially abundant features, as estimated from nominal count data, spanned 32% to 74% in these studies (see Table 4). This differential abundance was often small in absolute terms and demonstrably difficult to detect relative to the scale of sample-sample variation. Features which were significantly differentially abundant in the nominal count data often fell below the threshold of significance in the observed counts. An representative example of this from Kimmerling et al. is shown in S5 Fig.

Prediction of outcomes in these cases (Figs 5 and 6) was dichotomous. None of the observed sensitivities were within 50% intervals of model prediction. However, 100% of the observed specificities were within 50% predictive intervals for specificity.

**Moderate sensitivity and specificity cases.**   Most data sets exhibited intermediate sensitivities and specificities. Median sensitivity in this group, which included Hagai et al., Song et al., Monaco et al., Vieira-Silva et al., Grüen et al., Muraro et al., and Yu et al., was 0.47 and median specificity was 0.84. These data sets spanned a large range of estimated percent differentially abundant features in their nominal counts, from 18% to 79%, and mean fold change in total abundance across conditions ranged from 1.1 to 2. Count tables contained between 76 and over 26,000 features. Within-data set sensitivity and specificity were variable across methods. In some data sets, like those of the Monaco et al. and Vieira-Silva et al. studies, there is evidence of a trade-off in sensitivity and specificity (Fig 5).

Sensitivity prediction was poor again. Only 24% of observed sensitivities were within the 50% interval of prediction on sensitivity but 79% of specificities were within the same 50% predictive interval.

**High sensitivity, low specificity cases.**   A final group contained the data sets of Owens et al. and Klein et al., each of which had an overwhelming majority of differentially abundant features, as estimated from differences in nominal counts. These majorities of differentially abundant features contributed to similarly large, several-fold change in total abundance across conditions for each set of nominal counts. Experimentally, both involved studies of tissue differentiation—one in zebrafish, one in mouse. Though sensitivity was at its highest in this data, specificity was overall very low. We note that this is at least partially attributable to a reduction in the number of stably abundant features in these data sets, as false positive *rates* are a function of the ratio of "positive" and "negative" calls. For a complementary perspective in terms of false positive counts, see S6 Fig. Nevertheless, a representative false positive from the data of Klein et al. is shown in S5 Fig. The direction of change in this and other features is reversed in the "observed" count data relative to nominal counts, suggesting a compositional effect.

Accuracy of prediction was lower for sensitivity than for specificity in these data sets. The observed, high sensitivity fell within the 50% predictive interval in 90% of cases. However, none of the per-method specificity predictions for the data of Owens et al. or Klein et al. were within the 50% predictive interval.

Summarizing results for all twelve data sets, we note that sensitivity was greatly reduced in these real data sets relative to simulation and was more difficult to predict as well. Observed sensitivities were within the 50% interval of model-prediction only 29% of the time overall, versus 71% for specificity. In this small sample, sensitivity and specificity constitute a tradeoff with an association coefficient of -0.54 ($p < 0.0004$). The lowest specificities (or highest false positive rates) are associated with those data sets encoding the largest amount of change, both in terms of the number of changing features and the scale of the resulting change in total abundance. However, as in Fig 2, false positive *counts* are lower in these data sets than those of Song et al., Monaco et al., and Grüen et al. (see S6 Fig).

Finally, we note that superficially similar data sets (e.g. those of Kimmerling et al. and Owens et al.; see Table 4) can yield very different outcomes on the basis of differences in sample-sample variation.

## Mitigation strategies

We pursued two mitigation strategies which we believed could improve the accuracy of differential abundance estimates from relative counts: 1) increased stringency of differential abundance testing and 2) normalization against stable reference features with DESeq2.

In the first case we both increased the significance threshold for differentially abundant features (FDR ≤ 0.01) and enforced a fold change (effect size) threshold for differential features. In essence, we now ask about the agreement between absolute and rescaled relative data in terms of big, unambiguous differences. In this setting, median specificity was improved from 0.90 to 0.95 and from 0.63 to 0.85 in the subset of data sets with a majority of differentially abundant features. For full results, see Table 2 and compare Figs 1 and 2 with S7 and S8 Figs respectively.

In a second analysis, we briefly explored the effect of normalization by stable reference features. DESeq2 allows a user to specify a set of reference features against which samples will be normalized using a "control_genes" option. We utilized this feature in three settings: 1) normalizing simulated data sets against low-variance features identified in absolute counts, 2) normalizing real data sets against low-variance features identified in absolute counts, and 3) normalizing real data against a set of 13 "housekeeping genes" pulled from the literature including *gapdh* and *hprt1* [40–43].

Results derived from low-variance gene normalization show mostly modest improvements in specificity in both simulated and real data (S7 and S8 Tables; S9 Fig). For example, median specificity was increased by over 15% in data sets with an initial "high" FPR of greater than 10%. This improvement was progressive: real data sets with the worst specificities showed the greatest improvement, from 0.653 to 0.902 (Song et al.), 0.830 to 0.896 (Monaco et al.), and 0.853 to 0.906 (Muraro et al.; S8 Table). The housekeeping gene set was less effective and even increased discrepancy in two data sets, suggesting that, although normalization by control features can improve accuracy substantially, caution should be employed here: strong biological prior knowledge or independent validating measurements may be required if researchers wish to propose stable genes for sample normalization.

## Discussion

While the potential for compositional effects to drive differential abundance has repeatedly been described in the literature, uncertainty remains about the scope of this problem. Previous studies have shown that rates of false positive differential abundance calls can be high in certain settings [2, 19, 20]. Our results indicate that the problem is at least partially a function of the amount of change in the system under study and that differential abundance estimates from experiments characterizing extreme change across observed conditions are likely to be distorted. Both simulated and published experimental data contributed to this picture. Data sets with low fold change across conditions and a minority of differentially abundant features had high specificities (low false positive rates) and low false positive counts.

We found prediction of outcomes was possible in a limited sense. In simulated data, especially discrepant outcomes were roughly predictable by a few summary statistics derived from their relative abundances—low sensitivity from features which captured information about sparsity in the data set, and low specificity from estimates of the number of differentially abundant features.

Accuracy outcomes were less predictable in real data where poor predictive performance generally took the form of overprediction. Extreme cases were more difficult to predict: where sensitivity was extremely low, per-method models tended to overpredict it and where there was a tremendous number of differentially abundant features, specificity was overpredicted.

Some simple strategies offer avenues for mitigating the discrepancy between absolute and relative estimates of differential abundance. In our analysis, increasing the stringency of differential abundance calling through stricter significance and effect size thresholds improved accuracy significantly. That is, absolute and relative data is more often in agreement about features exhibiting large, unambiguous changes across conditions. Secondly, we observe that normalization by known stable features modestly improves accuracy when performed by DESeq2. Where these stable features are known or can be experimentally discovered, analyses would likely benefit from their use. Similarly, the use of external reference features like spike-ins has been validated in plate-based single-cell assays [13]. Though their application remains challenging on droplet-based platforms, they may still offer the best experimental means of reconciling the relative and absolute perspectives of change.

Lastly, it should be noted that the concerns motivating this and similar studies may be moot for some types of sequence count data. In particular, some single-cell platforms generate library sizes (i.e. total per-sample observed abundances) which are already roughly proportional to absolute abundances. This is in line a view that abundances in deeply sequenced UMI-barcoded single cells are likely to be a good proxy for absolute abundances [12, 14]. The effect of compositionality may be a minor concern under these circumstances.

## Materials & methods

We simulated general purpose molecular count data. These counts are interpretable as a variety of biological quantities, for example, transcript abundance in a cell or bacterial strain abundance in a microbial community. The simulated abundances undergo a sampling step intended to loosely replicate the process of measurement itself and, crucially, the normalization of total abundance across samples, giving a second set of count data. We refer to the first set of count data as "absolute" counts and the second, resampled set as relative or observed counts and explored the degree to which this loss of information about changes in total abundance alters the results of a mock differential abundance analysis by simulating a wide range of settings in our data, where key characteristics like complexity of composition (e.g. gene number) and fold change across simulated conditions varied widely.

### Simulation model

We designed a simulation framework to generate count data corresponding to two arbitrarily different conditions, denoted by superscripts in the equations below. First, for $p = 1, \ldots, P$ features in the first condition, a set of log mean abundances was drawn as

$$\theta_p^{(1)} \sim \mathrm{N}(m, S^2), \tag{1}$$

where hyperparameters $m$ and $S$ tune the mean and standard deviation of baseline log abundances. A correlation matrix was drawn as

$$\Omega \sim \text{Inverse-Wishart}(n, Q) \tag{2}$$

where scale matrix $Q$ was supplied as either the identity matrix (for a minority of simulations) or a dense correlation matrix with net positive elements. The matrix $\Omega$ is subsequently rescaled to a correlation matrix and used to draw correlated feature perturbations in a second

condition as

$$\theta_p^{(2)} \sim \text{MVN}(\theta_p^{(1)}, \Omega \cdot a) \tag{3}$$

where the hyperparameter $a$ exists in order to tune the overall scale of the correlated log perturbations. Mean abundances on the scale of sequence counts for each condition are calculated as

$$\gamma_p^{(1)} = \exp(\theta_p^{(1)}), \quad \gamma_p^{(2)} = \exp(\theta_p^{(2)}) \tag{4}$$

Differentially abundant features in some desired proportion, $c$, are obtained as follows: features are selected as differentially abundant with probability $c$. For those selected features only, the perturbed $\gamma_p^{(2)}$ serves as the mean abundance in the second condition; for all other features, the mean abundance in both the first and second conditions is given by $\gamma_p^{(1)}$. Let these new vectors be $\mu_p^{(1)}, \mu_p^{(2)}$. These represent mean the abundances of $P$ features in two conditions, some of which differ across conditions, others of which are identical. Replicates $i = 1, \ldots, 10$ are then generated for each condition as follows. A fixed dispersion parameter for absolute counts is defined as $d_{\text{abs}} = 1000$ and those counts are drawn as

$$y_{i,p}^{(1)} \sim \text{NegBinom}(\mu_p^{(1)} \cdot \delta, 1000) \tag{5}$$

where

$$\delta \sim \max(0.1, \text{N}(1, g)) \tag{6}$$

(Note that the dispersion parameter has been chosen such that the resulting counts are only barely overdispersed with respect to a Poisson.) The purpose of the truncated, per-sample multiplier $\delta$ is to re-scale all abundances within a given sample by some factor of approximately 1 but by increasing the scale of hyperparameter $g$, increasing replicate noise can be added within a condition. This process is repeated for the second condition to give a set of absolute counts $y_p^{(2)}$.

A new average observed total abundance (or library size) is randomly drawn as

$$u \sim \text{Unif}(5000, 2 \times 10^6) \tag{7}$$

Finally, observed abundances $z$ are generated through a multinomial resampling procedure similar to that of [10, 20, 23] which gives relative abundances as counts per million. Where $i$ and $k$ index different samples prior to resampling, $i'$ indexes the sample $i$ after resampling, and total counts for sample $i$ prior to resampling are given by $n_i = \Sigma y_i$, we have

$$z_{i'} \sim \text{Mult}(\pi_{i'} = y_i/n_i, n_{i'} = n_k) \tag{8}$$

where superscripts have been suppressed as this procedure is identical across simulated "conditions." The resulting $P$-length vector of counts for a given sample contains relative but not absolute abundance information. These vectors are collapsed into a $P \times 20$ count matrix containing 10 replicate samples for each of two simulated conditions. In order to evaluate the discrepancy of differential abundance calling on observed versus absolute counts, we apply differential abundance methods to count matrices $Z$ and $Y$ respectively and score the differences.

## Breadth of simulations

In order to generate simulations with a wide variety of characteristics, we swept in a grid over all our hyperparameters. Feature number $P$ was stepped through values 100, 1000, and 5000. A maximum feature number of 5000 was chosen as these simulations were computationally intensive and major trends had become apparent at that scale. The degree of feature correlation was encoded in five realizations of scale matrices $Q$, encoding fully independent features at one extreme and 50% strongly positively correlated features at the other extreme. Log mean abundance ($m$) and the log variance ($S$) were independently incremented through low to high values. Likewise, the average log perturbation size ($a$) was swept from low to high in five steps, as a proportion of log mean abundance.

Replicate noise $g$ varied from low to high in three steps. And finally, the proportion of differentially abundant features ranged across 20%, 30%, 50%, 70%, and 90%. Note that because many "perturbations" were very small, detectable differential abundance was generally only a fraction of the parameterized amount and most data sets contain a minority of differentially abundant features. Overall this 5625 simulated data sets were generated with almost continuous variation characteristics of interest.

We suggest that ranges of these parameter settings approximately represent different data modalities. We term the Microbial setting that with low to moderate feature number ($P \leq 1000$) and largest average perturbation, in accordance with a belief that bacterial communities are often simple (in terms of sequence variants with more than negligible abundance) and that they are highly variable even at short time scales [44].

We designate the Bulk Transcriptomic setting as that with the largest feature number ($P = 5000$) and having a lower average perturbation, the rationale being that transcriptomes sampled in aggregate over many cells are complex but largely stable compositions. Similarly, we define the intermediate Cell Transcriptomic setting, approximately representative of single-cell RNA-seq data, to comprise simulations with moderate to large feature numbers ($P \geq 1000$) and moderate perturbation sizes. These categories are intended as rough outlines and we note that within these settings the realized data varies in terms of 1) degree of feature correlation, 2) overall abundance, 3) (un)evenness of composition, and 4) within-condition variation in totals.

## Calling differential abundance

Five differential abundance calling methods were used in this study. These were ALDEx2 v1.22.0 [23], ANCOM-BC v1.0.5 [24], DESeq2 v1.30.1 [25], edgeR v3.32.1 [26], and scran v1.18.5 [27]. Each of these methods relies upon the use of a reference quantity to renormalize sample total abundances. ALDEx2, DESeq2, and edgeR rely on various (trimmed) estimates of the per-sample mean or median abundance for this reference quantity. ANCOM-BC uses a model-derived estimate of the "sampling fraction" of total abundance represented in an observed sample. scran's procedure learns and applies a condition- or cell type-specific normalization factor. For simplicity, we omit from consideration models which lack a rescaling. We also omit zero-inflation models. Although these are popular in single-cell mRNA sequencing data, the debate continues about whether these models are appropriate for these data [45, 46].

Finally, differential abundance calls made on observed counts must be evaluated against a reference in order to calculate discrepancy. For this reference, we used calls made by a negative binomial generalized linear model on absolute abundances as a pseudo-gold standard or "oracle" in all cases. Features with significantly different means according to the GLM were considered "true" instances of differential abundance. One caveat is that some of the discrepancy in calls between absolute and relative abundances will be due to differences in sensitivity between the models applied to the reference and "observed" data sets—i.e. between a stock NB GLM

and DESeq2. In all settings, true positive, true negative, false positive, and false negative calls were manually spot checked to verify disagreements between methods were generally unambiguous. Accuracy (in terms of sensitivity and specificity) was calculated between differential abundance calls made on absolute abundances and observed abundances. Unadjusted p-values were collected from all methods and multiple test correction applied via p.adjust in R using Benjamini-Hochberg method.

### Predictive modeling

In total, we trained 10 random forest models over 61 summary features, for each combination differential abundance calling method (ALDEx2, ANCOM-BC, DESeq2, edgeR, or scran) and accuracy measure (sensitivity or specificity). All such predictive models were fit with the randomForest package in R [47]. A random forest is an ensemble of decision trees and this tree-based approach was chosen because, while extensive, our simulations were not exhaustive. We anticipated that learning sets of decision rules might generalize well to unseen conditions, in particular feature numbers larger than those we explored in simulation. Feature importance was measured as "gain," or the relative increase in predictive accuracy achieved by the inclusion in the model of a given feature, as computed by the caret package in R [48].

Predictive models built from these summary features attempted to estimate sensitivity and specificity values explicitly. Details on these models are given in the Methods. All models were trained on 80% of the simulated data and their predictive accuracy was assessed on the reserved 20%.

### Real data processing

Publicly available data was downloaded from sources provided in the published materials for the studies cited. In general, relative abundances were converted to counts per million and sequences for external spike-ins extracted from these counts. Simple per-sample scaling factors were calculated from mean spike-in abundances and applied to the relative data to give "nominal" abundances.

In the inhibited and control cells of Klein et al., the expression of housekeeping gene *Gapdh* was used to normalize per-sample total abundance. In the 16S metagenomic data of Barlow et al. and Vieira-Silva et al., nominal abundances had already been estimated by the authors of those studies using quantitative PCR-based methods. In the single-cell expression data of Yu et al., $\log_2$ observed total abundances correlated well with $\log_2$ spike-in abundances and the observed data themselves were treated as true abundances. Relative abundances were derived by normalizing per-sample library sizes and scaling to give counts per million. Finally, in the coupled cell mass and expression measurements of Kimmerling et al., the estimated cell mass was used as scaling factor for observed expression to give nominal abundances.

Nominal and relative abundances were then filtered to exclude features (genes or bacterial sequence variants) present at an average abundance of less than a single count in either the nominal or relative count tables. This both reduced the size of the largest data sets—making them more computationally manageable—and reduced sparsity in the most extreme cases.

### Supporting information

**S1 Text. Additional details on simulated data sets and mitigation strategies.** This file contains supplemental methods and results detailing the characteristics of simulated data sets, describes variable importance determination in predictive models, and gives further details on the effects of mitigation strategies like control genes.
(PDF)

**S1 Table. Median and thresholded simulated specificity by feature number.** Specificity in simulated data, FDR ≤ 0.05. Per-method results are grouped and sorted by increasing feature number.
(PDF)

**S2 Table. Predictive feature list.** Features of observed abundances used to predict sensitivity and specificity outcomes.
(PDF)

**S3 Table. Features informative for the prediction of sensitivity.** Predictive features and their relative importance (as gain) in the prediction of sensitivity.
(PDF)

**S4 Table. Features informative for the prediction of specificity.** Predictive features and their relative importance (as gain) in the prediction of specificity.
(PDF)

**S5 Table. Observed sensitivities on real data sets.** Observed sensitivities on real data sets.
(PDF)

**S6 Table. Observed specificities on real data sets.** Observed specificities on real data sets.
(PDF)

**S7 Table. Real data sensitivity with control features.** DESeq2 sensitivity when using a variety of features as controls (via the control_genes argument). Sensitivity is improved when using a random set of low-variance features as references against which to rescale observed abundances. A selection of "housekeeping" genes used for the same purpose gives less improvement over baseline.
(PDF)

**S8 Table. Real data specificity with control features.** DESeq2 specificity when using a variety of features as controls (via the control_genes argument). Specificity is improved when using a random set of low-variance features as references against which to rescale observed abundances. A selection of "housekeeping" genes used for the same purpose gives less improvement or worsens specificity in the data sets of Monaco et al. and Grüen et al.
(PDF)

**S1 Fig. Characteristics of simulated data.** Distributions associated with three characteristics of the 5625 simulated data sets: **a)** percent differentially abundant features, **b)** fold change in total abundance across conditions, and **c)** percent zeros.
(TIF)

**S2 Fig. Asymmetry of simulated differential abundance.** The number, scale, and direction of differential abundance varied randomly in simulation, giving rise to both increases and decreases in total abundance across conditions.
(TIF)

**S3 Fig. Simulated sensitivity and specificity as a function of feature number and proportion of differentially abundant features.** Sensitivity and specificity for five differential abundance calling methods in terms of increasing feature number from **a)** 100 to **b)** 1000 to **c)** 5000 features.
(TIF)

**S4 Fig. Simulated sensitivity and specificity as a function of setting and fold change.** Sensitivity and specificity for five differential abundance calling methods in three experimental settings: **a)** Microbial, **b)** Bulk Transcriptomic, and **c)** Cell Transcriptomic settings. Data sets are labeled by fold change across conditions.
(TIF)

**S5 Fig. Example discrepant differential abundance calls.** Discrepant calls in nominal and relative abundances. **a)** A typical false negative result in data derived from the experiment of Kimmerling et al. This feature (the gene *H2-T3*) is significantly differentially abundant in the nominal abundances but not in estimates made from relative abundances by DESeq2. **b)** A typical false positive result in data from Klein et al. associated with gene *UNG*, tested by edgeR (with TMM normalization).
(TIF)

**S6 Fig. False positive counts as a function of percent differentially abundant features in real data.** Counts of false positives as a function of the percent of simulated differentially abundant features for five methods applied to 12 real data sets.
(TIF)

**S7 Fig. Simulated sensitivity and specificity as a function of feature number and fold change, FDR $\leq$ 0.01 plus effect size thresholding.** Sensitivity and specificity for five differential abundance calling methods in three experimental settings: **a)** Microbial, **b)** Bulk Transcriptomic, and **c)** Cell Transcriptomic settings. Data sets are labeled by proportion of differentially abundant features. Here, differential abundance calling is subject to greater stringency: FDR $\leq$ 0.01 and a fold change across conditions of at least 2. Median specificity is improved from 0.90 (FDR $\leq$ 0.05) to 0.95.
(TIF)

**S8 Fig. False positive counts as a function of percent differentially abundant features in simulation, FDR $\leq$ 0.01 plus effect size thresholding.** Counts of false positives as a function of the percent of simulated differentially abundant features for five methods with stringent differential abundance calling (FDR $\leq$ 0.01 and a fold change across conditions of at least 2). Columns segregate simulations with increasing numbers of features. Counts of false positives are reduced relative to FDR $\leq$ 0.05.
(TIF)

**S9 Fig. DESeq2 simulated accuracy with "control" genes.** Sensitivity and specificity for all simulated data sets using DESeq2 **a)** without and **b)** with "control genes." Per-columns settings from left to right are Microbial, Transcriptomic (center column), and Bulk Transcriptomic. **c)** Distribution of the change in sensitivity and **d)** specificity following the introduction of control genes. Sensitivity is largely unchanged. Specificity is generally improved.
(TIF)

## Acknowledgments

The authors would like to acknowledge conversations with Justin Silverman, Lawrence David, and Tim Reddy for useful comments.

## Author Contributions

**Conceptualization:** Kimberly E. Roche, Sayan Mukherjee.

**Data curation:** Kimberly E. Roche.

**Formal analysis:** Kimberly E. Roche.

**Funding acquisition:** Sayan Mukherjee.

**Investigation:** Kimberly E. Roche.

**Methodology:** Kimberly E. Roche.

**Project administration:** Kimberly E. Roche.

**Resources:** Kimberly E. Roche.

**Software:** Kimberly E. Roche.

**Supervision:** Sayan Mukherjee.

**Validation:** Kimberly E. Roche.

**Visualization:** Kimberly E. Roche.

**Writing – original draft:** Kimberly E. Roche.

**Writing – review & editing:** Kimberly E. Roche, Sayan Mukherjee.

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
