## [Decision Letter · Decision Letter 0]

12 Apr 2022

Dear Ms. Roche,

Thank you very much for submitting your manuscript "The accuracy of absolute differential abundance analysis from relative count data" for consideration at PLOS Computational Biology.

As with all papers reviewed by the journal, your manuscript was reviewed by members of the editorial board and by several independent reviewers. In light of the reviews (below this email), we would like to invite the resubmission of a significantly-revised version that takes into account the reviewers' comments.

More simulation data sets are needed as suggested by one of the reviewers for evaluating the performance of the proposed method.

We cannot make any decision about publication until we have seen the revised manuscript and your response to the reviewers' comments. Your revised manuscript is also likely to be sent to reviewers for further evaluation.

Sincerely,

Jinyan Li

Associate Editor

PLOS Computational Biology

Ilya Ioshikhes

Deputy Editor

PLOS Computational Biology

More simulation data sets are needed as suggested by one of the reviewers for evaluating the performance of the proposed method.

Reviewer's Responses to Questions

**Comments to the Authors:**

Reviewer #1: This paper describes an investigation into the use of relative abundance measurements and normalisation to infer absolute abundance. This would be of interest to several different fields of researchers all using sequencing data. It's really nice to see that this has been applied across sequencing applications covering RNA sequencing and 16S sequencing. The paper is well written and easy to follow, however I have a few recommendations, mostly minor editing.

Recommendations

- Box 1 is really informative, but should be referred to within the main body of text.

- Line 127 Supplemental figures remain as question marks.

- Line 192 Supplemental table remains as a question mark.

- Inconsistent use of capital letters for "transcriptomic setting" and "microbial setting" (see lines 148-152 for example).

- I think Line 209 should be referring to figure 3 and figure 4? Otherwise figure 3 is not referred to.

- Figure 7 is referred to in the text before figure 6.

- On the work with the real data, these have been split into three sets of studies when describing the results. The criteria for this split is unclear and should be explictly described.

- There is a description of the differences between the normalisation methods employed by DESeq2, scran and ALDEx2 in the methods but I think this needs to be moved to the introduction, alongside a stronger justification of the choice of software. edgeR is arguably used a lot more often for RNA sequencing studies - why was this not considered? scran has been designed for single cell data - is it appropriate to be using this for bulk RNA and 16S data? Justification of this is needed. It feels like you're actually primarily picking single cell analysis software and then using these with bulk RNA seq and 16s data - are you actually interested in single cell analysis or all three analysis methods?

- I think you should be mentioning that none of the software selected are designed for use with 16S data - all have been developed with RNA sequencing data in mind. Therefore it's not surprising that the microbiome data performs poorly. This could also provide an opportunity to discuss the adoption of these softwares in the 16S research community and whether it is suitable.

- Typo line 364 - "txhese" should be "these"

- Where 6000 simulations has been mentioned in the text, this shoud be changed to the actual number used - 5625.

- The methods describe the statistics underlying each piece of software but no details are given on what cut off has been employed for each software for describing significantly differentially abundant e.g. FDR < 0.05. This should be stated.

- Figure 7 - no description of what red and blue means?

- From the paper, you're suggesting that spike ins are needed for more accurate abundance calculations but you've under stated this in the discussion. More emphasis should be given to this point.

Reviewer #2: The authors present a large scale assessment of the sensitivity and specificity of statistical methods that attempt to detect differential features from count datasets, without being provided with information about changes in the true total abundance across samples. While previous papers have taken more unequivocal positions on this question, e.g. that certain classes of methods must be used and others can never be successfully applied to experiments producing matrices of counts, this work I think rightly considers how the violation of model assumptions may result in loss of accuracy in degree, and furthermore shows that to some extent the loss of accuracy can be predicted by dataset features. This can hopefully guide practitioners to choosing methods that are likely to be reliable given particular types and features of their data. For example, DESeq2 and ALDEx2 had a relatively high number (15-18%) of simulated datasets with very low specificity (<50%) for the simulation of microbiome data, while this is much less likely for simulated bulk RNA-seq data (1-2%). While different simulation settings will capture data attributes of different data sets, any particular one simulation setting doesn't tell a complete story. The important contribution from this work can be seen in Figures 1 and 2, where by sweeping across a broad range of simulation settings, one can easily see that proportion of differential features or extent of effect sizes results in loss of specificity across methods.

Major comments:

It was hard from the simulation model description to get a sense for the scale of fold changes in the various simulations. It would help to include a histogram or plot showing the quantiles of the true LFC distribution for various example simulated datasets. In particular the tails.

Were the simulated fold changes always symmetric? It seemed this way from the equation for theta^(2). I would recommend to have at least a few simulations of asymmetric fold changes as these should lead to even quicker degradation of performance for methods that are not provided with information about the true abundance scale, as they often target the middle of the distribution of observed sample-to-sample ratios.

I didn't see what was the threshold used for assessing a positive call (5%, 10%?).

DESeq2 since it was originally released has the option to provide informative features ("controlGenes") in order to estimate the true abundance scale per sample (in the estimateSizeFactors function, the first step in DE analysis). It may be useful to know in this context how a small amount of information may or may not help methods that accept this type of information (maybe other methods can as well). In simulations in which a set of stable features exist, how do results change if 10 or 20 of such stable features are provided to the methods for the purpose of scaling factor estimation? For the real data examples, what if housekeeping genes (Gapdh, etc.) are provided to methods that accept these stable reference features? Do results change qualitatively?

Minor comments:

Line 81 "residual variation in observed total abundances across samples is generally taken to be technical noise" - I would consider to refer to this as a technical bias, instead of "noise" which often connotes variance.

Line 121 "low complexity" - what does this refer to specifically?

Consideration - correlation of sequencing depth (total count) and the condition variable has been described as a pathological situation for differential abundance, in particular given that many null features may have zeros for the lower sequenced group. Was this explored in any of the simulations?

Line 364 "txhese"

Line 391 - DESeq2 rescales using a simple method, just the median (over genes) of the ratio of each sample's count to the geometric mean over all samples. No Bayesian formula are used in this step.

What versions of software were used?

-Michael Love

**Have the authors made all data and (if applicable) computational code underlying the findings in their manuscript fully available?**

Reviewer #1: Yes

Reviewer #2: Yes

PLOS authors have the option to publish the peer review history of their article (what does this mean?). If published, this will include your full peer review and any attached files.

Reviewer #1: **Yes: **Sophie Shaw

Reviewer #2: **Yes: **Michael Love
---

## [Decision Letter · Decision Letter 1]

7 Jun 2022

Dear Dr. Roche,

We are pleased to inform you that your manuscript 'The accuracy of absolute differential abundance analysis from relative count data' has been provisionally accepted for publication in PLOS Computational Biology.

Best regards,

Jinyan Li

Associate Editor

PLOS Computational Biology

Ilya Ioshikhes

Deputy Editor

PLOS Computational Biology

The reviewers' comments have been fully addressed, and the reviewers are satisfactory with the revised version of the manuscript. One of the reviewers made two minor notes which do not need further review but which are worth of consideration.

Reviewer's Responses to Questions

**Comments to the Authors:**

Reviewer #1: Thanks to the authors for the careful consideration of my suggestions in the original review. I can see that sufficient further experimentation and results have been included, and key sections have been re-worded. Thank you for this! I'd now happily accept this manuscript for publication.

Reviewer #2: The authors have addressed my previous concerns.

I have two minor notes, which do not need further review:

1. "Recent versions of DESeq2" -- `controlGenes` is not a recent feature, it has been part of DESeq2 since its publication in 2014.

2. In the revised text the authors note "modest improvements in specificity" from use of low variance features as putatively stable features. In my assessment of the revised work, for some of these datasets, the improvement is not that modest: e.g. Song et al which goes from 65% to 90% specificity, which is also the dataset with the lowest specificity. The second lowest also gets a good boost in specificity, e.g. 83% to 90%, the third with 85% to 91%. The ones with highest specificity have the lowest increase in specificity, but this is expected, as there is less of a problem to correct there. These to me seem not that modest but a pretty direct and substantial way to mitigate the problems the paper sets out to describe and quantify. Likewise comparing Supp Fig 9a to 9b, it seems a lot of the worst case simulations in terms of specificity, the performance is substantially improved. I'll leave this to the authors discretion.

**Have the authors made all data and (if applicable) computational code underlying the findings in their manuscript fully available?**

Reviewer #1: Yes

Reviewer #2: Yes

PLOS authors have the option to publish the peer review history of their article (what does this mean?). If published, this will include your full peer review and any attached files.

Reviewer #1: **Yes: **Sophie Shaw

Reviewer #2: **Yes: **Michael Love

---

## [Editor Report · Acceptance letter]

8 Jul 2022

PCOMPBIOL-D-22-00333R1 

The accuracy of absolute differential abundance analysis from relative count data

Dear Dr Roche,

I am pleased to inform you that your manuscript has been formally accepted for publication in PLOS Computational Biology. Your manuscript is now with our production department and you will be notified of the publication date in due course.

With kind regards,

Agnes Pap
